# Dissolution of a submarine carbonate platform by a submerged lake of acidic seawater

Matthew P. Humphreys[1], Erik H. Meesters[2], Henk de Haas[3], Szabina Karancz[1], Louise Delaigue[1], Karel Bakker[1], Gerard Duineveld[1], Siham de Goeyse[1], Andi Haas[4], Furu Mienis[1], Sharyn Ossebaar[1], and Fleur C. van Duyl[4]

[1]Department of Ocean Systems (OCS), NIOZ Royal Netherlands Institute for Sea Research, PO Box 59, 1790 AB Den Burg (Texel), the Netherlands
[2]Wageningen Marine Research, Wageningen University and Research, PO Box 57, 1780 AB Den Helder, the Netherlands
[3]National Marine Facilities (NMF), NIOZ Royal Netherlands Institute for Sea Research, PO Box 59, 1790 AB Den Burg (Texel), the Netherlands
[4]Department of Marine Microbiology and Biogeochemistry (MMB), NIOZ Royal Netherlands Institute for Sea Research, PO Box 59, 1790 AB Den Burg (Texel), the Netherlands

*Correspondence to*: Matthew P. Humphreys (matthew.humphreys@nioz.nl)

**Abstract.** Submarine sinkholes are found on carbonate platforms around the world. They are thought to form and grow when groundwater interactions generate conditions corrosive to carbonate minerals. Because their morphology can restrict mixing and water exchange, the effects of biogeochemical processes can accumulate such that the sinkhole water properties considerably diverge from the surrounding ocean. Studies of sinkhole waters can therefore reveal new insights into marine biogeochemical cycles, thus sinkholes can be considered as 'natural laboratories' where the response of marine ecosystems to environmental variations can be investigated. We conducted the first measurements in recently discovered sinkholes on Luymes Bank, part of Saba Bank in the Caribbean Netherlands. Our measurements revealed a plume of gas bubbles rising from the seafloor in one of the sinkholes, which contained a constrained body of dense, low-oxygen ($[O_2] = 60.2 \pm 2.6 \ \mu mol \cdot kg^{-1}$), acidic ($pH_T = 6.24 \pm 0.01$) seawater that we term the 'acid lake'. Here, we investigate the physical and biogeochemical processes that gave rise to and sustain the acid lake, the chemistry of which is dominated by the bubble plume. We determine the provenance and fate of the acid lake's waters, which we deduce must be continuously flowing through. We show that the acid lake is actively dissolving the carbonate platform, so the bubble plume may provide a novel mechanism for submarine sinkhole formation and growth. It is likely that the bubble plume is ephemeral and that other currently non-acidic sinkholes on Luymes Bank have previously experienced 'acid lake' phases. Conditions within the acid lake are too extreme to represent future marine environmental responses to anthropogenic $CO_2$ emissions on human timescales, but may reflect the impact of proposed schemes to mitigate climate change by the deliberate addition of $CO_2$ and/or alkalinity to seawater. Other Luymes Bank sinkholes do host conditions analogous to projections for the end of the 21st century and could provide a venue for studies on the impacts of anthropogenic $CO_2$ uptake by the ocean.

# 1 Introduction

Submarine sinkholes are voids in carbonate platforms that are open to the surrounding marine waters (Mylroie et al., 1995). Their formation and growth are usually driven by mixing between groundwaters of different salinity, which can generate conditions corrosive to the carbonate minerals that form the platform (Wigley and Plummer, 1976). In general, submarine sinkholes are thought to have formed during periods of low sea level (glacial lowstands), when their platform would have been subaerial with a meteoric freshwater supply, and subsequently submerged during postglacial sea level rise (Mylroie, 2013), although groundwater circulation at the edges of continental shelves can also generate new submarine sinkholes (Land et al., 1995). After the initial void has formed, seawater-filled sinkholes can grow further through carbonate dissolution, for example due to acidity generated by sulfate reduction (Bottrell et al., 1991) or organic matter remineralisation (Andersson et al., 2007). Despite being open to the surrounding ocean, sinkhole morphology can restrict water exchange, allowing the effects of such biogeochemical processes to accumulate. Consequently, sinkholes can provide new insights into how marine biogeochemical cycles operate under conditions that differ from the present-day bulk ocean (Yao et al., 2020; Qiao et al., 2020).

A number of submarine sinkholes were recently discovered on Luymes Bank (Fig. 1), a promontory of Saba Bank (van der Land, 1977). Saba Bank is a ~2200 km$^2$ submerged carbonate platform in the Caribbean Sea (17.5°N, 63.5°W), designated a Particularly Sensitive Sea Area (PSSA), and a hotspot of marine biodiversity (Hoetjes and Carpenter, 2010). Its origin is not well known, but it is most likely a composite volcanic island, overlain by a limestone cap several kilometres thick (Westermann and Kiel, 1961; Despretz et al., 1985; van Benthem et al., 2013) that hosts a mixture of slowly accumulating carbonate sediments (Macintyre et al., 1975). Luymes Bank may be geologically distinct from Saba Bank; it has been suggested that its igneous foundation belongs to the Lesser Antilles volcanic arc and is only connected to Saba Bank by more recent limestone deposits (Westermann and Kiel, 1961; Bouysse, 1984).

Volcanic islands and seamounts often host hydrothermal $CO_2$ seeps, where gases rise from the seafloor as a plume of bubbles. As gases dissolve out of the bubbles, they modify seawater biogeochemistry, again generating conditions that differ from the surrounding ocean. For example, submarine vents releasing $CO_2$ (but not associated with sinkholes) have been widely used as 'natural laboratories' representing both past conditions, for palaeo-proxy validation (e.g. Witkowski et al., 2019), and projected future conditions, for investigating the effects of ocean acidification on marine ecosystems (e.g. González-Delgado and Hernández, 2018; Aiuppa et al., 2021).

Here, we describe the biogeochemistry of the waters within the submarine sinkholes of Luymes Bank for the first time. Our main aim is to determine how the conditions observed in the sinkholes are maintained through interactions with off-platform seawater and in-sinkhole biogeochemical processes. We consider how these sinkholes may have formed and discuss the implications for the surrounding carbonate platform. Finally, we put the sinkhole waters in the context of past and future oceanic conditions. Our results provide the baseline understanding of how this unique marine ecosystem

functions, which is necessary for future studies that may use the sinkholes of Luymes Bank as a natural laboratory to shed light on the resilience and adaptation of life to a changing environment.

## 2 Materials and methods

### 2.1 Research expeditions

The data in this study come from two research cruises, both on RV *Pelagia*. Cruise 64PE432 was a round trip from Saint Martin, from 13th to 24th February 2018 (Duineveld and Mienis, 2018). This first cruise was focussed on the deep northern slope of Saba Bank and thus included relatively few observations of the Luymes Bank sinkholes. Cruise 64PE465 was a round trip from Guadeloupe, from 5th to 18th December 2019 (van Duyl and Meesters, 2019). This second cruise was focussed primarily on the Luymes Bank sinkholes and thus provided the majority of the data presented here.

### 2.2 Multibeam surveys

Multibeam bathymetry and water column data were collected during both cruises 64PE432 and 64PE465. The multibeam echo sounder was a Kongsberg EM302 with a beam opening angle of 1×2° (Tx×Rx), maximum swath width of 150°, and nominal frequency of 30 kHz. Position and attitude information came from a combined Seapath380 and MRU5, both from Kongsberg/Seatex. During 64PE432, the ping mode was set to auto and an FM pulse was used, to optimise for bathymetric data collection. During 64PE465, a CW pulse was used, to optimise for water column reflection recording. Multibeam data were processed and visualised using QPS Fledermaus software.

### 2.3 Hydrography

#### 2.3.1 Sensor measurements

Seawater was collected and measured during cruise 64PE465 (Fig. 1c). At each station, we deployed a rosette equipped with sensors (1 Hz resolution) for conductivity-temperature-depth (CTD; Seabird Scientific), dissolved oxygen ($[O_2]$), photosynthetically active radiation (PAR), and transmissivity (C-Star Transmissometer, Seabird Scientific, 25 cm optical path length), plus water sampling bottles (11 l) to collect seawater from various depths. The $[O_2]$ sensor data were recalibrated based on Winkler titrations (Winkler, 1888; Pai et al., 1993; Labasque et al., 2004) of 24 seawater samples collected during the cruise that covered the full range of $[O_2]$ values observed. All hydrographic data from this cruise are freely available online (Humphreys et al., 2021a).

$[O_2]$ was also measured by an optical sensor (SBE 63, Sea-Bird Scientific) deployed together with CTD sensors (SBE37-SMP, Sea-Bird Scientific) on a video camera frame during cruise 64PE432. Continuous measurements were collected by these sensors along a transect across sinkhole S (Fig. 1c) with the frame held at varying depth but always approximately 2 m above the seafloor.

Apparent oxygen utilisation (AOU) was determined for both cruises as the difference between saturated $[O_2]$ (i.e. its value in equilibrium with the atmosphere) and measured $[O_2]$, with the former calculated from temperature and salinity using the combined-fit parameterisation of García & Gordon (1992, 1993). To first order, AOU thus quantifies how much $O_2$ has been used up by biological processes within the ocean since waters last equilibrated with the atmosphere.

### 2.3.2 Marine carbonate system

Samples for dissolved inorganic carbon ($T_C$), total alkalinity ($A_T$) and pH were collected via silicone tubing into completely filled borosilicate glass bottles (Schott Duran) with ground glass stoppers, following best-practice recommendations (Dickson et al., 2007a). These samples were stored in the dark until analysis. The pH samples were additionally poisoned with 50 µl saturated mercuric chloride solution and a 2.5 ml air headspace was added before storing sealed with greased ground-glass stoppers to prevent gas exchange.

$T_C$ and $A_T$ were measured at sea with a VINDTA 3C (#17; Marianda, Germany) within 24 hours of sample collection. The measurements were calibrated against batch 171 certified reference material (CRM) from Prof A. G. Dickson (Scripps Institution of Oceanography, USA). $T_C$ was measured by coulometric titration (UIC Inc., IL, USA) of a ~20 ml subsample with excess phosphoric acid (Dickson et al., 2007b). Based on duplicate samples, $1\sigma$ precision for $T_C$ was 0.5 µmol·kg$^{-1}$ ($n = 40$ duplicate pairs) and the CRM measurements had a standard deviation of 1.9 µmol·kg$^{-1}$. $A_T$ was measured by potentiometric titration of a ~100 ml subsample with ~0.1 M hydrochloric acid (Dickson et al., 2007c). $A_T$ was calculated from the titration data by least-squares fitting using Calkulate v2.3.0 (Humphreys and Matthews, 2020). Based on duplicate samples, $1\sigma$ precision for $A_T$ was 1.1 µmol·kg$^{-1}$ ($n = 35$ duplicate pairs) and the CRM measurements had a standard deviation of 2.0 µmol·kg$^{-1}$.

The marine carbonate system was solved from $T_C$ and $A_T$ using PyCO2SYS v1.7.0 (Humphreys et al., 2021b, c), with the carbonic acid dissociation constants of Lueker et al. (2000), the bisulfate dissociation constant of Dickson (1990) and the total borate:chlorinity of Uppström (1974), to calculate pH on the Total scale (pH$_T$), seawater $CO_2$ fugacity ($fCO_2$), and saturation state with respect to calcite ($\Omega_c$) and aragonite ($\Omega_a$). Uncertainties in $T_C$ and $A_T$ measurements were propagated through these calculations together with the uncertainties for equilibrium constants and total borate:chlorinity estimated by Orr et al. (2018).

Seawater pH was also measured directly in August 2021 using an optode pH sensor (PHROBSC-PK7-SUB, PyroScience GmbH, Germany; nominal pH range 6–8, NBS scale). A 2-point calibration of the optode was performed against PyroScience pH buffers with pH$_{NBS}$ values of 2 and 11. The sensor was left in each sample to stabilise for 20 minutes while recording 2 measurements per minute. Temperature variations were automatically compensated for by a PyroScience Pt100 thermometer. To minimise pH changes caused by $CO_2$ degassing during the measurement, measurements were carried out in a glove bag inside which a micro-atmosphere matching each sample's calculated $pCO_2$ was created using a mixture of compressed air and $CO_2$. The $pCO_2$ of the air within the glove bag were monitored using a Vernier Go Direct® $CO_2$ Gas

Sensor. Measured pH, which was on the NBS scale, was then converted to the Total scale ($pH_T$) and to the in situ temperature using PyCO2SYS v1.7.0 (Humphreys et al., 2021b, c).

### 2.3.3 Nutrients

Nutrient samples were collected in 60 ml high-density polyethylene (HDPE) syringes connected to a three-way valve via tubing and were drawn directly from the water sampling bottles without any air contact. After sampling on deck, the samples were processed immediately in the lab. They were filtered over a combined 0.8/0.2 μm filter. The samples were analysed at most 10 hours after collection. Analyses were carried out using HDPE "pony vials" with a volume of 6 ml as sample cups fitting the auto-sampler. All pony-vials plus caps were rinsed three times with sample before filling. For analysis, all pony-vials were covered with Parafilm to avoid influx from ammonia and evaporation during the measurement. Analyses were carried out with Seal QuAAtro instruments, as follows.

Dissolved inorganic nitrate and nitrite ($[NO_3^-] + [NO_2^-]$) were measured by diazotation of nitrite with sulphanilamide and $N$-(1-naphtyl)ethylene diammonium dichloride to form a reddish-purple dye measured at 540 nm (Hansen and Koroleff, 1999). Nitrate was separately first reduced in a copperised Cd-coil using imidazole as a buffer and then measured as nitrite. The method detection limits were 0.012 μmol·l$^{-1}$ for $[NO_3^-] + [NO_2^-]$ and 0.002 μmol·l$^{-1}$ for $[NO_2^-]$ alone; the $1\sigma$ precision values were 0.5 % and 0.4 % respectively. Total ammonia ($T_{NH_3} = [NH_3] + [NH_4^+]$) was measured by formation of the indo-phenol blue complex by using phenol and sodium hypochlorite at pH ~10.5 (Helder and De Vries, 1979). Citrate was used as a buffer and complexant for calcium and magnesium. The colour was measured at 630 nm. The method detection limit for $T_{NH_3}$ was 0.09 μmol·l$^{-1}$ and the $1\sigma$ precision was 0.7 %. The sum of all the N-containing species is termed 'dissolved inorganic nitrogen' ($T_N = [NO_3^-] + [NO_2^-] + T_{NH_3}$).

Orthophosphate ($T_P$) was measured by formation of a blue reduced molybdenum phosphate-complex at pH 0.9–1.1 following Murphy and Riley (1962). Potassium antimonyl tartrate was used as a catalyst and ascorbic acid as a reducing reagent. The colour formed was measured at 880 nm. The method detection limit was 0.01 μmol·l$^{-1}$ and the $1\sigma$ precision was 0.8 %.

Dissolved silica ($T_{Si}$) was measured as a blue reduced silicate molybdenum-complex at 810 nm following Strickland and Parsons (1972). Ascorbic acid was used as reducing reagent and oxalic acid was used to prevent interference of phosphate. The method detection limit was 0.02 μmol·l$^{-1}$ and the $1\sigma$ precision was 0.2 %.

### 2.4 Attribution of sinkhole processes

To determine the effects of processes within or en route to the sinkholes on the chemistry of their waters, we need to know the waters' 'preformed' properties, that is, their properties before any sinkhole-related processes had acted on them. Differences between measured and preformed properties are thus termed the 'sinkhole effect'. For any property $P$ (e.g. temperature or $T_C$):

$$\Delta P = P_{\text{meas}} - P_{\text{pref}} \tag{1}$$

where $P_{\text{pref}}$ is the preformed value, $P_{\text{meas}}$ is the value measured within the sinkhole, and $\Delta P$ is the sinkhole effect.

We estimated $P_{\text{pref}}$ using data from off-platform sampling stations to either side of Luymes Bank. We fitted each property of interest (i.e. off-platform $\theta$, $[O_2]$, $T_C$, $T_{Si}$, $T_P$, $A_T$, $[NO_3^-]$, $[NO_2^-]$, and $T_{NH_3}$) to a polynomial or exponential function (whichever fit best) of each of salinity ($S$; Supp. Fig. S1) and potential density anomaly ($\sigma_0$; Supp. Fig. S2). The functions fitted to $S$ were used to determine $P_{\text{pref}}$ within the acid lake, defined as deeper than 187 m and within the area marked on Fig. 1c, while the functions fitted to $\sigma_0$ were used elsewhere. Explicitly, vertical profiles of $P_{\text{pref}}$, and thereby $\Delta P$ using Eq. (1), were determined by inputting the S and $\sigma_0$ values observed within the sinkholes into the relevant fitted functions.

$\Delta T_C$ was split into contributions from organic matter (OM) remineralisation ($\Delta T_C(OM)$), carbonate mineral dissolution ($\Delta T_C(CO_3)$) and a residual ($\Delta T_C(r)$), following the approach of Brewer (1978):

$$\Delta T_C = \Delta T_C(OM) + \Delta T_C(CO_3) + \Delta T_C(r) \tag{2}$$

$\Delta T_C(OM)$ was estimated from each of $\Delta T_N$, $\Delta T_P$ and $\Delta[O_2]$ using the OM stoichiometry of Anderson and Sarmiento (1994) (i.e. $R_{C/N} = 117/16$, $R_{C/P} = 117/1$, $R_{C/O_2} = -117/170$, where $R_{C/X}$ is the stoichiometric ratio between carbon and element $X$ during OM remineralisation). This stoichiometry, an update of the canonical ratios of Redfield et al. (1963), was determined by analysis of carbon, nutrient and oxygen gradients across multiple major ocean basins, and represents the average effect of biological activity (primary production and remineralisation) on these variables (Anderson and Sarmiento, 1994), assuming that these processes on average follow the reaction

$$117\,CO_2 + 16\,(NO_3^-,\ NO_2^-,\ NH_4^+) + PO_4^{3-} \rightleftharpoons OM + 170\,O_2 \tag{3}$$

where the forwards reaction represents primary production of OM, and the reverse its remineralisation; the brackets indicate that any one of the enclosed species could be used, without affecting our analysis. The estimates from each of $\Delta T_N$, $\Delta T_P$ and $\Delta[O_2]$ were consistent with each other (Sect. 3.3), so they were averaged:

$$\Delta T_C(OM) = \frac{1}{3}\left(R_{C/N}\,\Delta T_N + R_{C/P}\,\Delta T_P + R_{C/O_2}\,\Delta[O_2]\right) \tag{4}$$

$\Delta T_C(CO_3)$ was estimated from $\Delta A_T$, corrected for the effects of nutrient changes on $A_T$ following Wolf-Gladrow et al. (2007):

$$\Delta A_T(CaCO_3) = \Delta A_T + \Delta[NO_3^-] + \Delta[NO_2^-] - \Delta T_{NH_3} + \Delta P_T \tag{5}$$

$$\Delta T_C(CaCO_3) = \frac{1}{2}\Delta A_T(CaCO_3) \tag{6}$$

Finally, $\Delta T_C(r)$ was calculated by rearranging Eq. (2):

$$\Delta T_C(r) = \Delta T_C - \Delta T_C(OM) - \Delta T_C(CO_3) \tag{7}$$

$\Delta T_N$, $\Delta T_P$ and $\Delta[O_2]$ were directly proportional to each other and to $\Delta T_C(OM)$ (by definition; Eq. 4) in every sinkhole, including the acid lake. However, there were two distinct ratios between $\Delta T_{Si}$ and $\Delta T_C(OM)$: one in the non-acidic sinkholes and another in the acid lake. We assumed that the former ratio (in the non-acidic sinkholes) represented the stoichiometric coefficient for OM-associated silicate remineralisation ($R_{C/Si}$) and we determined its value by fitting $\Delta T_{Si}$ to

$\Delta T_{\mathrm{C}}(\mathrm{OM})$ using data from sinkholes N and S excluding the acid lake. Within the acid lake, fitted $R_{\mathrm{C/Si}}$ was then used to separate $\Delta T_{\mathrm{Si}}$ into the component associated with OM remineralisation ($\Delta T_{\mathrm{Si}}(\mathrm{OM})$) and a residual ($\Delta T_{\mathrm{Si}}(\mathrm{r})$):

$$\Delta T_{\mathrm{Si}}(\mathrm{r}) = \Delta T_{\mathrm{Si}} - \Delta T_{\mathrm{Si}}(\mathrm{OM}) \tag{8}$$

where

$$\Delta T_{\mathrm{Si}}(\mathrm{OM}) = R_{\mathrm{C/Si}}\, \Delta T_{\mathrm{C}}(\mathrm{OM}) \tag{9}$$

Therefore, $\Delta T_{\mathrm{Si}}(\mathrm{r})$ is by definition zero outside of the acid lake, where all of $\Delta T_{\mathrm{Si}}$ is attributed to $\Delta T_{\mathrm{Si}}(\mathrm{OM})$.

## 3 Results and discussion

### 3.1 Sinkhole distribution and morphology

Our multibeam data revealed tens of sinkholes across Luymes Bank. Towards the south there were many relatively small ($< 0.15\ \mathrm{km^2}$) and shallow ($< 67\ \mathrm{m}$ deeper than the platform) sinkholes, which are not discussed further here. In the north, there were four larger and deeper sinkholes, named N, S, E, and W (Fig. 1c). These might more accurately be termed sinkhole complexes, as their irregular shapes indicated that each consisted of several sinkholes merged together.

Sinkholes W and E were partially exposed at the edges of Luymes Bank, whereas sinkholes N and S were fully surrounded by the platform. The deepest lateral connection to open-ocean waters, termed the 'connection depth', was 93 m for sinkholes N and S, 142 m for W, and 166 m for E. Sinkholes N and S were separated from each other only beneath 187 m, which was the deepest point on an internal ridge (Fig. 1c). Water column multibeam reflections revealed a density contrast just beneath this depth in sinkhole N only (Fig. 1d). As will be shown, this density contrast marks the upper surface of the 'acid lake'. Multibeam data also showed a vertical plume of bubbles rising from the seafloor here (Fig. 1d). Other, much smaller bubble plumes were also identified elsewhere in sinkhole N, and in the northern part of sinkhole S.

Together, sinkholes N and S had a volume of 0.6766 $\mathrm{km^3}$ and surface area of 5.32 $\mathrm{km^2}$. Within this, the acid lake (i.e. sinkhole N beneath 187 m) contained 0.0882 $\mathrm{km^3}$ with a surface area of 1.38 $\mathrm{km^2}$.

### 3.2 Sinkhole water origins

We determined the preformed properties of the waters within the Luymes Bank sinkholes based on conservative properties, that is, properties that, beneath the mixed layer at the surface ocean, can be altered only through mixing. Salinity ($S$), potential temperature ($\theta$), and potential density anomaly ($\sigma_0$, which is strongly controlled by $S$ and $\theta$) are conservative in the open ocean, so they have been widely used as water mass tracers (Tomczak, 1999). In each sinkhole except the acid lake, the vertical profiles of $S$, $\theta$ and $\sigma_0$ followed open-ocean patterns (which were typical for the region; Morrison and Nowlin, 1982) down to a certain depth, beneath which they became relatively homogeneous (Fig. 2a–c). This 'certain depth' was different for each sinkhole, but in each case corresponded closely to the connection depth (Sect. 3.1). We deduce that the waters

within each sinkhole originated from the adjacent open ocean at the corresponding connection depth. Deeper, denser open-ocean waters were presumably blocked from entering the sinkholes by the platform's bathymetry.

While the upper layer of sinkhole N followed the pattern described above, the physical tracers ($S$, $\theta$ and $\sigma_0$) changed
considerably towards saltier, cooler and denser values across a transition zone from ~190 to ~220 m (Fig. 2a–c). This transition marked the surface of a constrained body of acidic ($pH_T$ = 6.24 ± 0.01), low-oxygen ($[O_2]$ = 60.2 ± 2.6 μmol·kg$^{-1}$) seawater in the sinkhole, which we term the 'acid lake'. In the acid lake, the average value of each physical tracer ($S$, $\theta$ and $\sigma_0$) corresponded to a different off-platform water depth: $S$ corresponded to ~260 m (Fig. 2a), but $\theta$ (Fig. 2b) and $\sigma_0$ (Fig. 2c) corresponded to 190–200 m. For these waters to have a common off-platform source, some process must have caused either
a decrease in $S$ by ~0.45 or an increase in $\theta$ by 2.0 °C (Table 1). As bubble plumes arising from hydrothermal activity are often associated with seawater heating (Aiuppa et al., 2021), but not with freshening, we assumed $S$ was unaffected by sinkhole processes and used it as the conservative tracer to define preformed properties. This indicated an open-ocean provenance at ~260 m (Fig. 2a), roughly the same depth at which the bubble plume emerges (Fig. 1d). The acid lake's waters were far denser than overlying waters in sinkhole N, and too dense to have travelled over the platform. We infer that these
open-ocean waters must instead have reached the acid lake through channels through the limestone platform.

**3.3 Waters within the acid lake**

In addition to changes in $\theta$ and $\sigma_0$, many biogeochemical properties of the acid lake had been substantially altered from their preformed states (Table 1). In terms of the marine carbonate system, the seawater $pH_T$ at 6.24 ± 0.01 (Fig. 2d) was about 1.8 lower than in the preformed open-ocean waters – a ~63-fold increase in $[H^+]$ – due to a ~2.4-fold increase in $T_C$ (Table 1).
Despite the unusual composition of acid lake waters, we are confident in the calculated $pH_T$ values thanks to their excellent agreement with the direct optode measurements (Supp. Fig. S3). The seawater $fCO_2$ was 49800 ± 1200 μatm, and $\Omega_c$ and $\Omega_a$ were both less than 0.14; when $\Omega < 1$, the corresponding mineral is unstable and is expected to spontaneously dissolve (Morse et al., 2007). $A_T$ was elevated in the acid lake, which has roughly the opposite effect than elevated $T_C$ on pH, $fCO_2$ and $\Omega$ (Humphreys et al., 2018), but the $A_T$ increase by a factor of ~1.5 (Table 1) was insufficient to fully compensate $\Delta T_C$,
so $\Delta T_C$ dominated the net impact on the carbonate system.

We used Eqs. (2) to (7) to unravel the different drivers of $\Delta T_C$. $\Delta T_C$(OM) was determined from changes in nutrients and $[O_2]$. All measured nutrient concentrations were considerably elevated from preformed values in the acid lake, and $[O_2]$ was lowered by ~61 % (Table 1). $\Delta T_C$(OM) can be calculated from any one of $\Delta T_N$, $\Delta T_P$ or $\Delta[O_2]$ and the relevant stoichiometric ratio to carbon, but as we found that all three options gave consistent results with the OM stoichiometry of
Anderson and Sarmiento (1994) (Fig. 2e), we took their mean (Eq. 4). The consistency between using $[O_2]$ and the nutrients means that no processes beyond OM remineralisation need be invoked to explain the $[O_2]$ distribution in the acid lake. $\Delta T_C$(OM) represented ~5 % of the total $\Delta T_C$ (Fig. 2f). It also suggests that the OM stoichiometry of Anderson and Sarmiento (1994) is accurate for this ecosystem, despite more recent studies highlighting regional variability in the elemental ratios (Martiny et al., 2013).

$\Delta T_{Si}$ behaved differently from the other nutrients. Except in the acid lake, it had a constant stoichiometric coefficient ($R_{C/Si}$; Eq. 9) of 15.4. Although not strictly OM, silicate minerals are closely associated with OM when production is dominated by siliceous plankton such as diatoms (Tréguer et al., 2021), and our calculated $R_{C/Si}$ is consistent with this scenario (Brzezinski, 1985). However, applying this $R_{C/Si}$ to the acid lake (Eq. 8) revealed an excess of $T_{Si}$ beyond that expected from OM remineralisation ($\Delta T_{Si}(r)$; Fig. 2g). $\Delta T_{Si}(r)$ was uniformly greatest in a deep sub-layer beneath

~240 m within the acid lake, which was also identifiable as a step to lower transmissivity (Fig. 2g). $\Delta T_{Si}(r)$ could have come from sinking OM, with its excess indicating that a portion of the other remineralised nutrients had been taken up by non-siliceous organisms within the acid lake. Alternatively, $T_{Si}$ might have been released from dissolution of siliceous particles such as sponge spicules from within the limestone platform (Bertolino et al., 2017; Maldonado et al., 2021), or supplied in the hydrothermal fluids themselves (Mortlock et al., 1993; Tréguer and De La Rocha, 2013).

Returning to $\Delta T_C$, $\Delta T_C(CO_3)$ was determined from changes in $A_T$ and nutrients. We interpret $\Delta A_T$ as resulting mainly from carbonate mineral dissolution, modulated by the relatively minor effect of changes in nutrients (Eq. 5). $\Delta A_T(CO_3)$ corresponds to 4693 ± 17 t of $CaCO_3$ being dissolved into the acid lake. Assuming a bulk density of 1.55–2.75 g·cm$^{-3}$, this corresponds to 1700–3000 m$^3$ of limestone in solution, which is 0.002–0.003 % of the acid lake's volume. $\Delta A_T(CO_3)$ was related to $\Delta T_C$ (Eq. 6) to reveal that $\Delta T_C(CO_3)$ represented ~20 % of the total $\Delta T_C$ in the acid lake (Fig. 2f).

$\Delta T_C(r)$ therefore represented ~75 % of total $\Delta T_C$ (Fig. 2f). We interpret $\Delta T_C(r)$ as arising from $CO_2$ dissolving out from the gases of the bubble plume that we observed (Fig. 1d).

## 3.4 Water flow through the acid lake

The density contrast across the submarine surface of the acid lake was visible on multibeam data. Similar reflections just above the seafloor on the southeast flank of the ridge between sinkholes N and S indicated that dense acid-lake waters might

be flowing out over the ridge (Fig. 1d). Further evidence for an overflow came from sensor data indicating sharp decreases in $[O_2]$ (Fig. 2h) and changes in other variables towards acid-lake values. Therefore, overflow into sinkhole S appears to have limited the vertical extent of the acid lake, which was only found beneath the deepest point on the ridge (~187 m).

An outflow must be balanced by an inflow. Further evidence for ongoing inflow came from $[O_2]$: while low, it was far from anoxic, as would be expected in isolated waters (Garman et al., 2011; Xie et al., 2019). The acid lake was too dark

for photosynthesis, with the euphotic zone over the sinkholes only 80 ± 7 m deep (based on a criterion of 1 % of surface PAR; Lee et al., 2007). The higher density of acid lake waters likely prevents mixing with overlying waters from being an $O_2$ source throughout the acid lake. We speculate that active transport of oxygenated open-ocean seawater through the platform may be driven or enhanced by tidal pumping (Martin et al., 2012; van Haren et al., 2019), or by a Venturi effect as the bubble plume intersects seawater-filled channels.

However, we did not find the distinctive hydrographic signature of acid lake waters elsewhere in sinkhole S, even at its deepest point. The fate of the overflowing waters thus remains unclear. Mixing within sinkhole S, possibly enhanced as the acid-lake waters flow down the rugged flank of the ridge, might prevent the denser waters from forming into a permanent

layer. Furthermore, it appears that sinkhole S is sporadically refreshed with off-platform waters. Near-seafloor sensor measurements across sinkhole S in February 2018 found a consistent AOU of ~80 μmol·kg$^{-1}$, but measurements in the same

sinkhole in December 2019 found a significantly lower deep maximum AOU of ~55 μmol·kg$^{-1}$. The decline in AOU indicates that sinkhole S was at least partly replenished with off-platform waters between the cruises, which could further dilute acid-lake waters. Alternatively, the dense waters could drain out through other caves or fissures. A third possibility is that the acid lake is a recent feature, not yet present for long enough for its overflow to build up in sinkhole S.

### 3.5 Impact and wider context

Low pH and $\Omega$ show that carbonate mineral dissolution is possible in the acid lake, elevated $A_T$ shows that dissolution has happened, and a continuous through-flow of waters shows that dissolution is ongoing. Together, this suggests an alternative mechanism for submarine sinkhole formation that does not rely on mixing between groundwaters of different salinity, as previously thought essential (Mylroie et al., 1995; Land et al., 1995). Instead, the $CO_2$-rich geothermal fluids are sufficiently acidic to independently dissolve bedrock and create new submarine sinkholes. We cannot be sure which process was

responsible for initially creating the sinkholes of Luymes Bank, which would have been exposed to meteoric freshwater during glacial lowstands (e.g. ~21.5 ka ago, when sea level was ~130 m lower; Lambeck et al., 2014). But it is remarkable that, despite recent extensive multibeam surveys, no sinkholes have been found elsewhere on Saba Bank, which is mostly shallower than Luymes Bank. This observation is consistent with the Luymes Bank sinkholes having been created by $CO_2$ seeps, especially if its volcanic basement is indeed a separate geological entity from Saba Bank (Westermann and Kiel,

1961; Bouysse, 1984) and if only the former currently gives rise to bubble plumes.

The morphology of the acid lake, being wide relative to its depth, and with a gradual rounded slope between its floor and walls, was quite different from the taller, thinner shapes observed in some other marine sinkholes (e.g. Li et al., 2018), but it was similar to the other Luymes Bank sinkholes, suggesting they had been shaped by similar processes. However, seawater was supersaturated with respect to calcite and aragonite in sinkhole S and in N above the acid lake,

indicating that carbonate minerals were not actively dissolving (unless within sediment porewaters). In other words, if submarine dissolution has been an important control on the acid lake's morphology, then the other sinkholes may have previously experienced an (or several) acid-lake phase(s). The acid lake's bubble plume is therefore probably ephemeral and others could appear elsewhere on Luymes Bank, with changes perhaps associated with tectonic and seismic activity in the underlying bedrock. Indeed, there was multibeam evidence for several smaller bubble plumes within sinkhole N and at the

northern end of sinkhole S, but we are not able to distinguish their biogeochemical impact from that of the main bubble plume.

Besides explaining the acid lake, an important motivator for this study was to quantify the conditions inside the Luymes Bank sinkholes for their potential future use as 'natural laboratories' to investigate marine ecosystem responses to past or future environmental change. Taking each property individually, the acid lake did represent certain periods in Earth's

history: in terms of [$O_2$], it was analogous to the average global near-surface ocean from 1.85 to 0.85 Ga before present

(Holland, 2006); seawater $fCO_2$ was also analogous to the Precambrian surface ocean, while $pH_T$ was lower than that modelled even during the Archaean, 4.0 Ga before present (Krissansen-Totton et al., 2018). However, taken as a complete ecosystem, the acid lake is not truly representative of this ancient environment, due to both its ephemeral nature and the many other ways that the Earth system has changed in the intervening time. Looking to the future, $fCO_2$ and pH in the acid

lake were too extreme for projected ocean acidification due to passive ocean uptake of anthropogenic $CO_2$ (Caldeira and Wickett, 2003), and such an interpretation would be further complicated by the low $[O_2]$. But the acid lake could represent the aftermath of controlled addition of $T_C$ and/or $A_T$ to the ocean, which has been proposed as a geoengineering technique to counteract anthropogenic climate change (Adams and Caldeira, 2008; Gattuso et al., 2018). Indeed, using variations in density to ensure added $CO_2$ remains deep within the ocean and does not escape back into the atmosphere has been a key

feature of some suggested schemes, such as adding $T_C$ to naturally dense water masses (Marchetti, 1977) or even using pure liquid $CO_2$, which is denser than seawater and stable under the pressure of the deep ocean (Goldthorpe, 2017). The high-$T_C$ waters of the acid lake were similarly trapped by the density contrast at its surface; studying how effective a barrier to $CO_2$ exchange this density contrast is and investigating the consequences of the unusual conditions for the ecosystems both within and around the acid lake could give new insights into the efficacy and impacts of these potential climate mitigation schemes.

Sinkhole S could represent anthropogenic ocean acidification, with less extreme $fCO_2$ and pH values similar to projections for 2100 under a pessimistic, business-as-usual emission scenario (Caldeira and Wickett, 2003). However, given the large changes in AOU we observed over a 2-year interval (Sect. 3.4), we would need to better understand how variable conditions in this sinkhole are before it could be relied on as a natural laboratory.

**4 Conclusion**

Luymes Bank is a submarine carbonate platform hosting a series of sinkholes, including one partly filled by an 'acid lake' of dense, low-oxygen seawater. These acidic waters may have originated in the open ocean, travelled through channels in the platform, and had a considerable amount of dissolved $CO_2$ added to them from a gas bubble plume arising from the seafloor. The extra $CO_2$ dominated the seawater chemistry of the acid lake, with lesser effects from organic matter remineralisation and dissolution of the surrounding limestone platform, the latter of which may have provided the mechanism for sinkhole

formation and growth here. The morphologies of the non-acidic sinkholes were similar to that of the acid lake, suggesting that they may have previously experienced acid-lake phases, thus the acid lake's bubble plume may be ephemeral. Observed overflow of the acid lake's waters over a ridge and into an adjacent sinkhole indicated an ongoing supply of open ocean waters to the acid lake. The fate of the overflowing waters is unclear, but their presence in the adjacent sinkhole gave rise to conditions akin to end-of-century projections for the global mean surface ocean. However, interannual variations in

dissolved $O_2$ indicated that the mixing between acid lake and open ocean endmembers in the adjacent sinkhole was not in steady state, so more work is needed to better characterise the stability and longevity of this unique ecosystem before it can

be relied upon as a natural laboratory. We expect such further study of the sinkholes of Luymes Bank to deliver new insights into how marine ecosystems and biogeochemical cycles may adapt to profound environmental change.

## Data availability

The hydrographic and biogeochemical data presented here, together with the cruise report (van Duyl and Meesters, 2019), are freely available online at https://doi.org/10.25850/nioz/7b.b.yc.

## Author contributions

Conceptualisation: M.P.H., E.H.M., A.H., H.H., S.K., F.M., K.B., S.G., G.D. and F.C.D. Data curation: M.P.H. and H.H. Investigation: M.P.H., H.H., S.K., L.D., K.B., S.G. and S.O. Methodology, Software, Visualisation, and Writing – original
draft preparation: M.P.H. Writing – review and editing: M.P.H., E.H.M., A.H., H.H., S.K., L.D., F.M., K.B. and F.C.D.

## Competing interests

The authors declare that they have no conflict of interest.

## Acknowledgements

We are grateful to the officers and crew of RV *Pelagia* and technical support from the NMF department before for their
support and assistance during our research expeditions. We thank Lennart de Nooijer, Olivier Sulpis, John Pohlman, and two anonymous reviewers for their feedback, which greatly improved this manuscript. Cruise 64PE432 was carried out within the framework of NICO (Netherlands Initiative Changing Oceans).

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

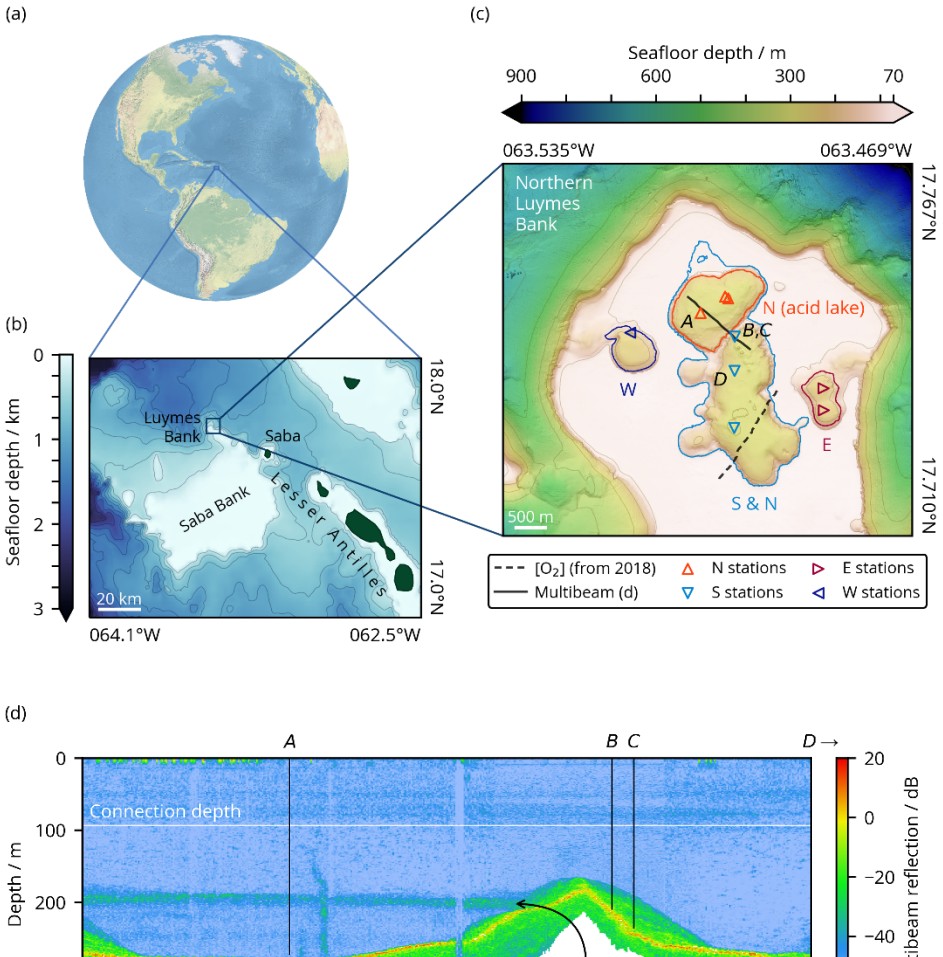

Figure 1: Location of Luymes Bank (a) in the Caribbean/North Atlantic and (b) relative to Saba Bank and the Leeward Islands of the Lesser Antilles (GEBCO Compilation Group, 2020), with contours at 250 m depth intervals corresponding to colour-bar ticks. (c) Bathymetry of northern Luymes Bank showing our sampling stations (triangles) and sensor transects (lines). Sinkholes N (containing the acid lake), S, E and W are enclosed by the coloured lines, which are contours at each sinkhole's connection depth (noting that this is the same for S and the non-acid part of N, labelled 'S & N'; Sect. 3.1), and can also be identified from the orientation and colour of the station markers within them (see legend). Contours are at 100 m depth intervals plus at 70 m, matching the colour-bar ticks. Sampling stations labelled *A*, *B*, *C* and *D* correspond to the same labels in (d) and in Fig. 2h. (d) Water column reflectivity (multibeam) transect across the ridge between sinkholes N (acid lake) and S, marked with a solid line in (c). The horizontal white line shows the connection depth to the open ocean (93 m). Vertical black lines illustrate the maximum depth of measurements at each station (Fig. 2h).

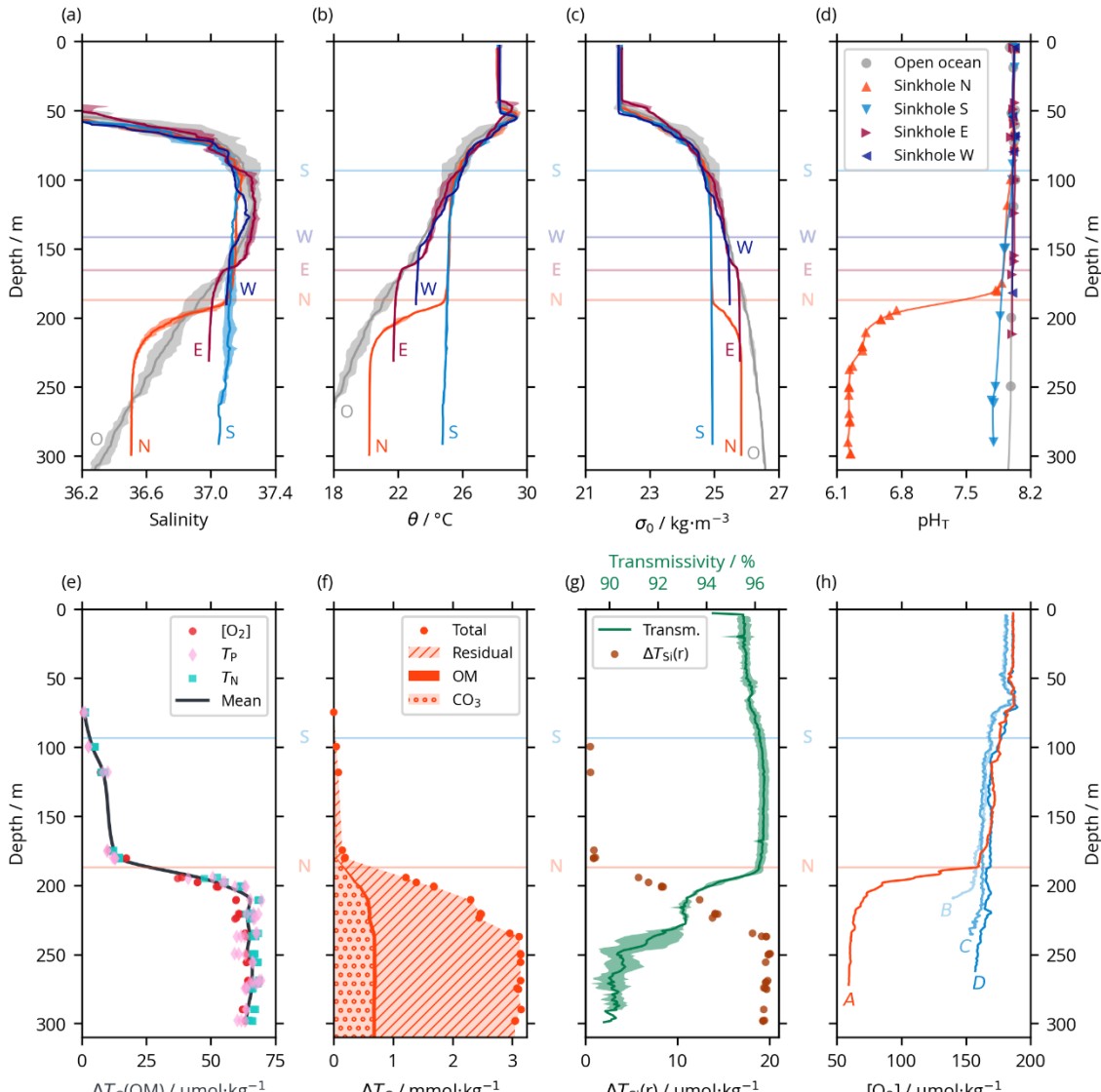


**Figure 2. Vertical profiles of selected hydrographic and biogeochemical properties of seawater observed within the Luymes Bank sinkholes and in nearby open-ocean waters. Data from all stations marked on Fig. 1c are shown together; the different stations are not distinguished, except in (h). Horizontal lines show the connection depths to off-platform waters (for sinkholes W, E, and S, where S also applies to the upper, non-acidic layer of sinkhole N) or the 'internal' connection between sinkholes S and N (labelled**
**N); (e)–(h) do not include data from sinkholes E and W, so their connection depths are not shown. (a) Practical salinity, including open-ocean waters in grey (labelled 'O'). Where multiple profiles were measured, shaded areas show ±1 standard deviation (SD) of the mean. (b) Potential temperature ($\theta$), styled as in (a). (c) Potential density ($\sigma_0$), styled as in (a). (d) Calculated $pH_T$, in all sinkholes and in the nearby open ocean. Uncertainties are smaller than the width of the markers. (e) $\Delta T_C(OM)$ in the acid lake, estimated from each of $\Delta[O_2]$, $\Delta T_P$ and $\Delta T_N$, which are consistent with each other. (f) Total $\Delta T_C$ in the acid lake is dominated by the**
**residual term, with smaller contributions from carbonate mineral dissolution and OM remineralisation. (g) A two-layer structure within the acid lake can be seen in some (but not all) variables, including transmissivity (line; shaded area indicates ±1 SD from the mean for multiple profiles) and $\Delta T_{Si}(r)$ (points). (h) [O₂] profiles in sinkholes N and S and on the flank of the ridge that**

separates them reveal acid lake waters overflowing into sinkhole S. The locations of profiles *A*, *B*, *C* and *D* are marked on Fig. 1c–d.


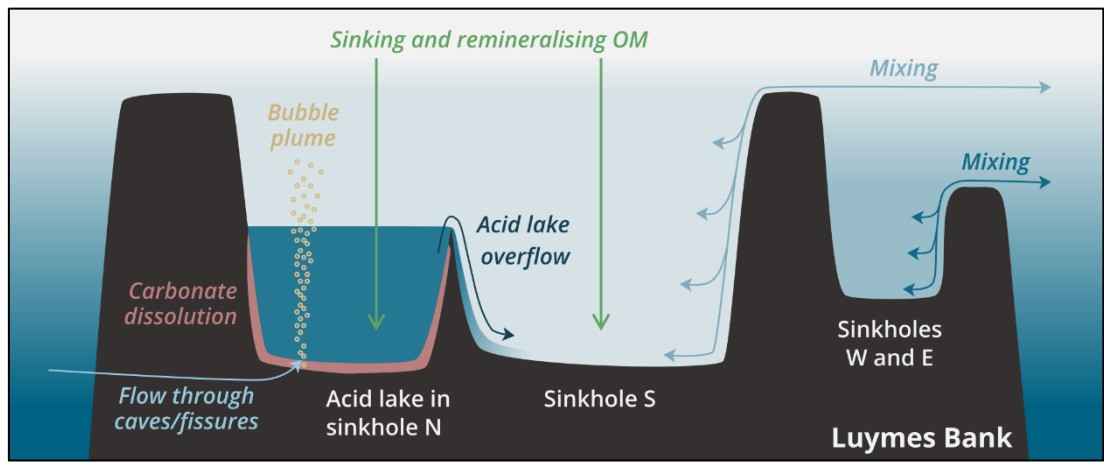

**Figure 3. Schematic cross-section of Luymes Bank with overview of the processes acting on seawater as it flows through the sinkholes. Depth range shown is from 0 to ~350 m. Background colour represents seawater density, increasing from light to dark blue.**


**Table 1. Maximum sinkhole effects on acid lake seawater properties.**

| Property | Acid lake extremum | Sinkhole effect at extremum | Unit |
|---|---|---|---|
| Practical salinity | 36.50 | 0* | |
| Potential temperature ($\theta$) | 20.2 | +2.0 | °C |
| Dissolved oxygen ($[O_2]$) | 60.0 | −93.0 | $\mu mol \cdot kg^{-1}$ |
| Dissolved inorganic carbon ($T_C$) | 5.27 | +3.10 | $mmol \cdot kg^{-1}$ |
| Total alkalinity ($A_T$) | 3.69 | +1.30 | $mmol \cdot kg^{-1}$ |
| Dissolved inorganic nitrogen ($T_N$) | 17.4 | +9.2 | $\mu mol \cdot kg^{-1}$ |
| Dissolved inorganic phosphate ($T_P$) | 0.97 | +0.56 | $\mu mol \cdot kg^{-1}$ |
| Dissolved inorganic silicate ($T_{Si}$) | 30.5 | +28.1 | $\mu mol \cdot kg^{-1}$ |

*By definition (we assumed that salinity in the acid lake had not been affected by sinkhole processes).