# Peer review of "Dissolution of a submarine carbonate platform by a submerged lake of acidic seawater"

_Biogeosciences, 2021_

## Author Comment (AC1)

**Reviewer 1**

Humphreys et al., characterize biogeochemical (dissolved) properties in sink holes, recently discovered in the Caribbean. The data is high quality and they make use of it to infer local processes, how sink holes may have evolved and for what purposes their study may be useful in the future. Their study is novel, interesting, and well written. I enjoyed reviewing this paper and have only few comments (mostly related to readability and clarifications).
**Thank you for your positive assessment of our work.**

General comment:
- I got confused quite often with when data referred to the sinkhole and when did it refer to the acid lake. For example, T_Si seems to refer to the acid lake but it is introduced in the methods to refer to the sinkhole. If possible, this should be carefully assessed in the revised version and cleaned up.

**The acid lake totally fills one of the sinkholes (up to a certain depth), so there is indeed some real overlap here. In the case of T_Si, we have revised our explanation of what this means in the Methods (see related responses below), which we hope makes this easier to follow.**

Comments along the text:

Introduction:
The introduction was well written and referenced, and was informative for a person who has no specific knowledge on sinkholes like me.

Methods:
The calculation of AOU should be described at least briefly.
**We updated this sentence to read: "Apparent oxygen utilisation (AOU) was determined for both cruises as the difference between saturated $[O_2]$ (i.e. in equilibrium with the atmosphere) and measured $[O_2]$, with the former calculated from temperature and salinity using the combined-fit parameterisation of García & Gordon (1992, 1993)."**

Where was the optical O2 sensor mounted? Outside the vessel at the surface? Or on the CTD? This was confusing and also did not become obvious from Fig. 1c. Please clarify.
**We updated this sentence to read: "$[O_2]$ was also measured by an optical sensor (SBE 63, Sea-Bird Scientific) deployed together with CTD sensors (SBE37-SMP, Sea-Bird Scientific) on a video camera frame during cruise 64PE432. Continuous measurements were collected by these sensors along a transect across sinkhole S with the frame held approximately 2 m above the seafloor (Fig. 1c)."**

I would abbreviate dissolved inorganic carbon as DIC as it seems most widely used. Or otherwise C_t, following Dickson et al. (2007). But I don't have strong feelings about this.
**We prefer to stick with $T_C$ for consistency with the notation for the other nutrients and the PyCO2SYS manuscript (Humphreys et al., Geosci Model Dev, in press).**

Since the constants by Sulpis et al. are relatively new, it would be useful to provide a comment why Humphreys et al. use those the most for their calculations.

*The constants of Sulpis et al. are virtually identical to the well-established and widely used Lueker et al. constants at the temperatures considered in this study; Sulpis et al. adjusted the Lueker et al. equations so that they fit overdetermined measurements from GLODAP and SOCAT better at the lowest temperatures. We have therefore switched to using the Lueker et al. constants instead, as we are working in the temperature range where the Sulpis et al. constants are no different, and this makes no difference to our results.*

I did not fully understand how the authors calculated the preformed values. When I look into the supplement then the S or sigma values in the sinkhole are a range. But for P_pref, I believe they need a distinct value. So what value would that be within the range? Is it the corresponding S or sigma value at the surface of the sinkhole? Please clarify and perhaps amend supplementary figures accordingly.

*There is not a single value for $P_{pref}$; it is calculated separately at each sampling depth within each sinkhole, hence the ranges of values seen on these figures. We have rephrased this paragraph, hopefully it is now clearer: "We estimated $P_{pref}$ using data from off-platform sampling stations to either side of Luymes Bank. We fitted each property of interest (i.e. off-platform $\vartheta$, $[O_2]$, $T_C$, $T_{Si}$, $T_P$, $A_T$, $[NO_3^-]$, $[NO_2^-]$, and $T_{NH_3}$) to a polynomial or exponential function (whichever fit best) of each of salinity (S; Supp. Fig. S1) and potential density anomaly ($\sigma_0$; Supp. Fig. S2). The functions fitted to S were used to determine $P_{pref}$ within the acid lake, defined as deeper than 187 m and within the area marked on Fig. 1c, while the functions fitted to $\sigma_0$ were used elsewhere. Explicitly, vertical profiles of $P_{pref}$, and thereby $\Delta P$ using Eq. (1), were determined by inputting the S and $\sigma_0$ values observed within the sinkholes into the relevant fitted functions."*

The description of the delta T_C components is not sufficient to understand what happened here (at least not for me). What is R_N, R_P, etc.? I assume R_N is the molar C/N ratio? This needs to be clarified.

*We added: "where $R_X$ is the stoichiometric ratio between carbon and element X during OM remineralisation".*

Also, for R_P there is only one number which is understandable if it is C/P but I would add the 1 anyways to avoid confusion.

*We added the 1.*

Related to this but more generally: I wonder how much sense it makes to use average Anderson and Sarmiento stoichiometry for this when we know that stoichiometry systematically varies across latitudes (or with nutrient concentration), according to Martiny et al., 2013/2014. It would perhaps be helpful to accommodate for this discussion. Shouldn't make much of a difference because T_C(OM) is low but it may raise some eye-brows.

*We added to Section 3.3: "[The consistency between OM remineralisation estimated from the different nutrients] suggests that the OM stoichiometry of Anderson and Sarmiento (1994) is accurate for this ecosystem, despite more recent studies highlighting regional variability in the elemental ratios (Martiny et al., 2013)."*

Line 164: This is unclear. Do you mean that delta TC(R) was delta TC-(deltaTC(OM)+deltaTC(CO3)) ? Perhaps just make a new equation to avoid confusion.

*Yes, we added a new equation as suggested.*

Line 165: what is the "stoichiometric coefficient for OM-associated silicate remineralisation (RSi)"?? Please try to be less cryptic and more descriptive. You lost me here until line 170. Please explain why you did, what you did here and wat the goal was before you provide how you did it.

***We updated this paragraph to better explain what we were doing here and why: "$\Delta T_N$, $\Delta T_P$ and $\Delta[O_2]$ were directly proportional to each other and to $\Delta T_C(OM)$ (by definition; Eq. 3) in every sinkhole, including the acid lake. However, there were two distinct ratios between $\Delta T_{Si}$ and $\Delta T_C(OM)$: one in the non-acidic sinkholes and another in the acid lake. We assumed that the former ratio (in the non-acidic sinkholes) represented the stoichiometric coefficient for OM-associated silicate remineralisation ($R_{C/Si}$) and we determined its value by fitting $\Delta T_{Si}$ to $\Delta T_C(OM)$ using data from sinkholes N and S excluding the acid lake. Within the acid lake, fitted $R_{C/Si}$ was then used to separate $\Delta T_{Si}$ into the component associated with OM remineralisation ($\Delta T_{Si}(OM)$) and a residual ($\Delta T_{Si}(r)$). [equations] Therefore, $\Delta T_{Si}(r)$ is by definition zero outside of the acid lake, where all of $\Delta T_{Si}$ is attributed to $\Delta T_{Si}(OM)$."***

Line 194: Perhaps remind the reader what the connection depth was.
***We added a pointer back to Section 3.1 where this is defined.***

Line 195: I am not sure about this statement. Do the authors mean that denser waters from deeper ocean regions cannot reach to the top of the platform because the surface of the platform is too shallow? If so, can the authors exclude that water may be pushed upwards somehow (e.g. upwelling) and then overflow into the sinkholes? I would agree that something like this seems unlikely but not sure if they can exclude this.
***We know that deeper, denser waters do not upwell and flow over the platform into the sinkholes because these sinkholes do not contain any denser waters. We added 'presumably' to this sentence to show that we cannot exclude that this ever happens. We also cannot envision any way that the denser waters of the acid lake could have flowed over the platform to get into the acid lake while not appearing in any of the other sinkholes.***

Line 197: check the unit for O2.
***Fixed, thank you.***

Line 198: I don't understand this sentence. Please explain why each physical tracer corresponded to a "significantly different" off-platform water depth.
***We updated the sentence, adding what these depths are, and references to new panels in Fig. 2 to help clarify what we mean here: "In the acid lake, the average value of each physical tracer (S, $\vartheta$ and $\sigma_0$) corresponded to a different off-platform water depth: S corresponded to ~260 m (Fig. 2a), but $\vartheta$ (Fig. 2b) and $\sigma_0$ (Fig. 2c) corresponded to 190–200 m."***

Line 205: I assume the authors have also considered that the corrosive conditions in the acid lake may have increased dissolution/chemical weathering which may have increased salinity? Can this thought be easily dismissed or is it worth mentioning it? The +0.45 in S cannot come from CaCO3 dissolution, I guess, but could it be something else? Do you have

data on major ion composition? (I am not sure if this is at all relevant but your data made me wonder so it may be worth throwing a sentence at this).

*The 0.45 number is how much salinity would need to have been decreased by through sinkholes processes if we were to assume that there had been no warming of the waters and so used a shallower off-platform preformed water depth for the acid lake. The acid lake is fresher than expected, not saltier. Dissolution of CaCO3 would act to increase salinity, so the change would be in the wrong direction.*

*We note that the differences in all biogeochemical properties between acid lake and off-platform waters were far greater than changes in these same properties with depth in the off-platform waters, so even if some processes had changed salinity within the acid lake and thus moved the preformed water depth up or down by tens of metres, it would make no meaningful difference to our analysis and interpretation.*

*We do not have major ion data but plan to investigate this in subsequent expeditions. Lacking this evidence, we prefer not to speculate on this further in this manuscript.*

Line 215: It is interesting that TA is comparatively little increased relative to DIC. Does this mean that the water inside the acid lake has a short residence time because it gets enriched with CO2 but doesn't have enough time to stay in there to dissolve CaCO3 to its full potential?

*This is indeed one possibility. Alternatively, dissolution within the limestone sediments may lead to a 'buffer zone' in the sediments with elevated alkalinity and Ω, preventing further dissolution.*

*As with the previous point above (on line 205), we plan to investigate this in subsequent expeditions, as we were unequipped to collect the required samples during the expedition described here, and as we have no real evidence yet either way, we prefer not to speculate on this here.*

Line 220: This should be moved to the methods, as I was wondering where the 1/3 of each was justified in equation 3. Or at least say that the reason for this will be given in secion XXX.

*In the Methods we changed the line before Eq. (3) from "The estimates were averaged:" to "The estimates were consistent with each other (Sect. 3.3), so they were averaged:"*

Fig. 2e. Why is delta T_si (R) only shown for sinkhole N, which has the acid lake (or is it?). The data representation is a bit confusing.

*We hope that the better explanation of what TSi(r) is in the updated methods (see our response to your point on our line 165 above) makes this clearer now: there is no residual excess Si anywhere apart from the acid lake, by definition.*

Line 286: I agree that the sinkholes could be useful natural analogues but I don't think the authors have provided particularly useful examples. The analogy to Precambrian conditions does not really make sense because that would mean that the acid lake is isolated for very, very long. But this does not seem to be the case as the authors state themselves. Thus, I don't think it has value as an analogue for that. The acid lake certainly has value for Ocean Acidification but care must be taken due to low O2, which may restrict its value. I think the acid lake constitutes an excellent natural analogue for the deliberate sequestration of CO2

into the ocean, either linked to "Direct Air Capture with carbon capture & storage (DACCS)", "Bio-energy with carbon capture & storage (BECCS)", or electrochemical splitting of $H_2O$ into NaOH and HCl (the "SEA MATE" approach from Eisaman et al), where the HCl could be sequestered in the deep ocean. I believe that the sinkholes have more values for this than what the authors mentioned so far. It may be worth considering this, potentially even in the abstract where they mention this also.

***We agree and we thank the reviewer for this insightful suggestion. We could not find any peer-reviewed publications on SEA MATE specifically but found other similar studies that were appropriate to illustrate the point.***

***We updated the sentence in the Abstract as follows: "Conditions within the acid lake are too extreme to represent the future environmental response to anthropogenic $CO_2$ emissions on human timescales, but may reflect the impact of proposed schemes to mitigate climate change by the deliberate addition of $CO_2$ and/or alkalinity to seawater."***

***We updated the Discussion paragraph to the following: "Besides explaining the acid lake, an important motivator for this study was to quantify the conditions inside the Luymes Bank sinkholes for their potential future use as 'natural laboratories' to investigate marine ecosystem responses to past or future environmental change. Taking each property individually, the acid lake did represent certain periods in Earth's history: in terms of $[O_2]$, it was analogous to the average global near-surface ocean from 1.85 to 0.85 Ga before present (Holland, 2006); seawater $fCO_2$ was also analogous to the Precambrian surface ocean, while $pH_T$ was lower than that modelled even during the Archaean, 4.0 Ga before present (Krissansen-Totton et al., 2018). However, taken as a complete ecosystem, the acid lake is not truly representative of this ancient environment, due to both its ephemeral nature and the many other ways that the Earth system has changed in the intervening time. Looking to the future, $fCO_2$ and pH in the acid lake were too extreme for projected ocean acidification due to passive ocean uptake of anthropogenic $CO_2$ (Caldeira and Wickett, 2003), and such an interpretation would be further complicated by the low $[O_2]$. But the acid lake could represent the aftermath of controlled addition of $T_C$ and/or $A_T$ to the ocean, which has been proposed as a geoengineering technique to counteract anthropogenic climate change (Adams and Caldeira, 2008; Gattuso et al., 2018). Indeed, using variations in density to ensure added $CO_2$ remains deep within the ocean and does not escape back into the atmosphere has been a key feature of some suggested schemes, such as adding $T_C$ to naturally dense water masses (Marchetti, 1977) or even using pure liquid $CO_2$, which is denser than seawater and stable under the pressure of the deep ocean (Goldthorpe, 2017). The high-$T_C$ waters of the acid lake were similarly trapped by the density contrast at its surface; studying how effective a barrier to $CO_2$ exchange this density contrast is and investigating the consequences of the unusual conditions for the ecosystems both within and around the acid lake could give new insights into the efficacy and impacts of these potential climate mitigation schemes."***

Line 299: "Likely"? Or is "may have" more appropriate?
***Switched to 'may have'.***

The Figures are generally really nice.
***Thank you!***

Fig. 1. Why are is one sinkhole (e.g. sinkhole S) at different locations? This is confusing.
***All the stations labelled S fall within sinkhole S (and for the other sinkholes/stations). We added contours that enclose each sinkhole to explicitly show their outlines.***

Fig. 2. Subplot b: Fig. 1 shows multiple sinkhole locations for sinkhole S, so what do the blue triangles in the case of sinkhole S refer to? (Same question for sinkhole N and E).
***The points shown are from all stations within each sinkhole; the different stations are not distinguished on these plots. We updated a sentence in the caption to clarify: "Data from all stations marked on Fig. 1c are shown together; the different stations are not distinguished, except in (h)."***

Table 1: Why are DIC and TA in mmol/kg when they are usually reported in µmol/kg? Not really important but distracted me a bit while reading.
***Normally µmol/kg is used because the variations of interest in open-ocean DIC and TA can be on the scale of 10 µmol/kg or smaller. However, here we are interested in variations on the scale of several thousand µmol/kg, so the mmol/kg unit is more convenient.***

**Reviewer 2**

Submarine sinkholes on carbonate platforms occur worldwide and have been recently discovered on Luymes Bank, Carribean Sea. Humphrey et al. study four different sinkhole complexes on Luymes bank – one of those sinkholes contains dense, acidic and oxygen-poor waters ('acid lake'). The authors study the formation of the sinkholes and of the 'acid lake' and decipher the physical and biogeochemical conditions and processes that sustain its occurrence using bathymetry and water column data.

I found the manuscript overall very well written and very well described. I felt confident that the authors have carried out their study carefully and thoroughly. Therefore, I only have minor comments.
***Thank you for this positive assessment.***

Minor comments:
I agree that sinkholes are useful 'natural laboratories' but I think that the connection the authors draw between the acid lake and Precambrian times is not really applicable as the time and spatial scales are very different – the authors themselves write that the bubble plume is likely only short-lived. Furthermore, many processes on Luymes Bank (e.g. difference in AOU) have not been studied in detail, and thus, the "Impact and wider context" should be addressed more carefully.
***We agree. This matches with a comment from the other reviewer, please see our response above (in reference to line 286).***

Line 52: Is there more data/knowledge on the occurrence and composition of hydrothermal seeps on Luymes Bank/Saba Bank, potentially also with regards to the host rock composition?
***We have described already what is known about the geology of Luymes Bank/Saba Bank, which is not very well understood. We are not aware of any previous data/knowledge about the seeps themselves, but we hope to investigate them further in subsequent expeditions.***

Line 92: Please extent on the calculation of AOU and for what it is used.
***The calculation of AOU was elaborated in response to the other reviewer. We also added, "To first order, AOU quantifies how much $O_2$ has been used up by biological processes within the ocean."***

Line 94: Is $T_c$ = DIC? I find the abbreviation $T_c$ confusing and I would recommend to re-name it to DIC.
***We prefer to stick with $T_C$ for consistency with the notation for the other nutrients and the PyCO2SYS manuscript (Humphreys et al., Geosci Model Dev, in press). DIC is fine within text but not appropriate as a term in mathematical equations following the IUPAC Green Book.***

Line 110: Please introduce $pH_T$.
***Changed to, 'pH on the Total scale ($pH_T$)'.***

Line 110: Maybe my knowledge is too limited but I am wondering what the difference/relation is between $pH_T$ and $pH_{NBS}$? How do you convert these?
***This is mentioned at the end of Section 2.3.2, i.e., 'using PyCO2SYS'. Please refer to that manuscript for details of the conversion.***

2.3.3 Nutrients: Please provide additional information on the data quality.
***We added method detection limits and precision values.***

Line 144: I have troubles understanding what is meant here with "preformed properties upon departure from the open ocean" – are those the off-platform seawater conditions right before the water enters the sinkhole? Please consider to rephrase.
***Rephrased to "their properties before any sinkhole-related processes had acted on them."***

Line 147: Please define/indicate, which $P_{pref}$ values are used for this calculation as Fig. S1-S2 only show the ranges. Maybe the $P_{pref}$ values can be added to Table 1 or indicated more clearly in Fig. S1â  S2.
***This section has been reworded in response to the other reviewer and is now hopefully clearer. Ppref is not a single value for each sinkhole, rather it was calculated separately at each depth, so there is not a single value that can be put into the table.***

Line 158: I would encourage the authors to give more detail on why the values have been set like this. Did Anderson and Sarmiento work in the same area, also shouldn't Refield be mentioned here?
***We added the explanatory sentence: "This stoichiometry, an update of the canonical ratios of Redfield et al. (1963), was determined by analysis of carbon, nutrient and oxygen gradients across multiple major ocean basins, and represents the average effect of biological activity (primary production and remineralisation) on these variables (Anderson and Sarmiento, 1994)."***

It took me quite long to understand that $R_N$ is actually C/N and $R_P$ = C/P. Why is "R" used here as this is otherwise introduced to represent the residual fractions? This is quite confusing.
*We agree that our notation was confusing. We switched to using $R_{C/N}$, $R_{C/P}$ etc. for the stoichiometric ratios, and a lowercase (r) for the residuals.*

Line 191: Why are the depth profiles for θ and σ not shown?
*We were trying to not overwhelm the reader with too many panels on this figure and thought that salinity alone communicated the physical part of the story the best. However, on reflection we think these would be useful, so we have added them to Figure 2 and the associated text.*

Line 199: "In the acid lake, each physical tracer…" maybe better "the (conc.) range of each physical tracer…"?
*Changed to 'the average value of each physical tracer'.*

Figure S1-S2: How did you determine such low NO2- concentrations? What is the LOQ of your method? Also for NH3 and Si?
*This info is now provided in the Methods section 2.3.3.*

It would be useful to have the abbreviations of the variables (such as $A_T$, $T_C$…) defined in the figure captions.
*We added these definitions to the figure captions in the SI.*

**Reviewer 3**

The study by Humphreys et all is a geochemical investigation of water masses in several sinkholes on the Saba Bank. The authors seek to identify the process that control the chemistry of the waters and relate those to the formation of the sinkhole. There is a lot going on with respect to where the waters originate, how they are altered during their migration through the platform and how waters move from one sinkhole to the other. Multibeam surveys are used to identify gas seeps and density layers. Sensors characterize the physicochemical properties of the water masses, and the analysis of total inorganic carbon, alkalinity, nutrients and silica are carefully integrated to attribute the sources of carbon and dissolution dynamics. Beyond being able to explain how material is transported and altered locally, the key contribution of this study is that they describe a novel mechanism for sinkhole creation that does not involve mixing or meteoric and seawater or sea-level oscillations. In this case, a geologic source of CO2 is entering the bottom of one of the sinkholes that creates a corrosive solution. The evidence in support of that is very convincing, and their speculation that this may have formed other sinkholes in the area reasonable. To what extent this process is important on a larger scale is unknown. I also appreciate the significance of finding a body of low pH water that may be used an analog of a more acidic ocean. Indeed this might be a great natural laboratory for such studies.
*Thank you for the positive assessment.*

The paper is well written and thoughtful. With only a few suggested edits and comments, this paper is ready to go. One general recommendation is that calling the deeper portion of

Sinkhole N an "acid lake" may be an exaggeration. While a pH of 6.4 is technically acidic, the acidity is very mild.
*We agree that the acidity is mild but as noted by the reviewer it is indeed acidic, and relative to the range of pH values normally experienced in the ocean it is extremely acidic, so we prefer to keep the terminology as it is.*

Specific Comments and Edits:

Line 14: The sentence about how sinkholes are thought to form does not include the importance of sea-level low stands. A slight edit to incorporate the effect of that process would be helpful.
*This is already mentioned in the relevant part of the Introduction. We tried to add it here in the Abstract but we found it requires quite a lot of additional information to explain the reasoning for why they are important, which breaks the flow there – the surrounding sentences are all about submarine sinkholes. Further, the low stand formation mechanism does not necessarily apply here, with the Luymes sinkholes potentially formed in a different way (from the bubble plume). Hence, we prefer to leave this as it is.*

Line 16: I would have "effects" to "products." I don't think effects accumulate. Products do. Sounds better to me at least.
*We prefer effects. Effects are changes due to a certain process and they can accumulate. Moreover, some of the sinkhole effects involve loss, not production (e.g. dissolved oxygen).*

Line 17: Just say the studies provide insight...aren't all insights new? It sounds less grandiose.
*By definition, not all insights are necessarily new, so we prefer to leave as is.*

Line 157: Consider adding the OM respiration equation so the stoichiometric ratios are more palpable.
*Great suggestion, thank you. We have added this (new Eq. 3).*

Line 185: Express the volume of the sinkhole as m3 rather than km3. I think most people can visualize that better.
*The numbers would be $6.766 \times 10^8$ $m^3$ and $8.82 \times 10^7$ $m^3$. Because of the $10^x$ terms, we are not convinced that these are easier to visualise than $0.6766$ $km^3$ and $0.0882$ $km^3$ (imagine a cube with kilometre sides, then take two-thirds of it for the first volume or about a tenth of it for the second), so we prefer to leave as is.*

I do not think much more than that needs to be said. This is an excellent paper that I think a lot of people will enjoy reading. Thank you for the opportunity to review it.
*Thank you for these kind words and for taking the time to read and review our paper. We greatly appreciate your suggestions and carefully considered them all, even though we have decided not made changes in several cases.*

---

## Author Response (AR2)

1. Thank you for raising the question of whether our Fig. 1d is accessible to colour-blind readers – we agree that this is an essential consideration. We put the figure through a simulator and we think that all the important features are clearly visible so the figure can be correctly interpreted:

[Figure]

We also checked that the figure caption and discussion in the main text do not refer to specific colours.

2. The data DOI now appears to work. The data repository was temporarily offline for maintenance when it was checked.

3. Figure 1 is original and our own work.

Figure 1a includes part of an image from Natural Earth Data, which may be freely reproduced without citation: https://www.naturalearthdata.com/about/terms-of-use/.

Figure 1b uses data from GEBCO, which is correctly cited in the caption.

Figures 1c and 1d include only our own data.